# Strengthening Timber Structural Members with CFRP and GFRP: A State-of-the-Art Review

**DOI:** 10.3390/polym14122381

**Published:** 2022-06-12

**Authors:** Khaled Saad, András Lengyel

**Affiliations:** Department of Structural Mechanics, Budapest University of Technology and Economics, H-1111 Budapest, Hungary; khaled.saad@emk.bme.hu

**Keywords:** timber, CFRP, GFRP, knot, bond characteristics, EBR, GiR, NSM, material models, FEM

## Abstract

The application of fibre-reinforced polymers (FRP) for strengthening timber structures has proven its efficiency in enhancing load-bearing capacity and, in some cases, the stiffness of structural elements, thus providing cost-effective and competitive alternatives both in new design and retrofitting existing historical buildings. Over the last few decades, several reinforcing materials and techniques evolved, and considerable progress was made in numerical modelling, especially using the finite element method. As this field of research has become extensive and diversified, as well as numerous contradicting results have emerged, a thorough review is necessary. This manuscript covers the topics of historical preliminaries, reinforcing with carbon and glass fibre composites, bond characteristics, main reinforcing techniques, modelling of knots, and the effects of the fibre waviness on the composite behaviour. A detailed overview is given on the experimental and numerical investigation of mechanics of strengthened beams. A one-of-a-kind table is presented that compares the stiffness improvement observed in several studies with analytical estimates. Attention is drawn to a number of challenges that have arisen, e.g., the moderate stiffness enhancement, composite-to-wood interface, modelling of knots, and strengthening of defected timber members. This paper can be used as a starting point for future research and engineering projects.

## 1. Aim and Scope of Pursued Study

The use of wood has grown significantly, and timber is now a popular building material for a wide range of exceptionally lightweight structures. Its ease of production and exceptional physical and mechanical characteristics are attributed to its low density and attractive landscapes. It is evident that, due to environmental concerns and its low energy demand, timber will be the most often utilized structural material.

Because of its orthotropic natural properties, timber is a complex material with limited analytical techniques to describe its basic behaviour. The existence of knots, splits, and grain slope also has a significant impact on the mechanical behaviour of a timber structural member, particularly when positioned in the tension zone. The material characteristics of wood might differ even among the same wood species, and many parameters are necessary for a complete model description. The behaviour of the tested specimens may vary, limiting the applicability of the results.

Fibre-reinforced polymer (FRP) materials can be synthetic composites manufactured from high-strength fibres by chemical synthesis (typically glass, carbon, aramid and basalt) [1,2], natural fibre-reinforced polymer composites (NFRP), or hybrid fibres (a mix of synthetic and natural fibres) [3,4] embedded in an adhesive matrix (usually epoxy, Polyester, Polyolefin). Synthetic reinforcing elements are available in a range of shapes and sizes. Very high strength and stiffness are almost the most important and common advantages of the synthetic fibre-reinforced composite materials. Usually, these mechanical properties are expressed in relative terms by dividing them by the density. Specific strength, density, or stiffness are also expressed by the obtained strength-to-density and stiffness-to-density quotients, those are especially preferable in weight-sensitive structures such as aircraft. The most challenging part of designing a composite material for a specific application is determining its composition. In almost all cases, mixing reinforcement with matrix changes the mechanical and physical properties of the resulting material, so it is critical to keep in mind that the microstructure of the material may change due to the presence of the reinforcement. Additionally, residual stresses may appear during manufacturing due to the difference in thermal expansion and conductivity. NFRP can be made of plant fibres, animal fibres, and mineral fibres, which can be easily found in nature. NFRP composites appear to have several benefits compared to synthetic fibres, including lower weight, price, toxicity, pollution, and recyclability [5].

A promising type of hybrid carbon/glass fibre plates were developed recently by [4,6]. It was demonstrated that the random fibre hybrid configuration may completely promote the synergetic effects of carbon and glass fibres. The obtained composite showed a good carbon/glass fibre/resin interface behaviour. However, the mechanical characteristics of hybrid plates were affected by increased exposure temperature and flexural loading level.

The interface failure was studied by [7] under different loads and finally, guidelines for the characterization were given. Many studies have also been conducted to investigate the simultaneous moulding of the composites and the wood adhesive [8].

It is frequently more cost-effective and less time-consuming to repair or reinforce damaged wood pieces rather than replace them entirely. FRP composite materials provide several benefits, including high elastic modulus, strength, and corrosion resistance. Carbon fibre-reinforced polymers (CFRP) have many advantages when it comes to strengthening wood. They are durable, stick to wood easily, and have low density. CFRP composite is a brittle material. However, when fitted to the tension face of a timber element under bending, a considerable amount of tensile stresses are transferred from timber allowing wood compression fibres to yield and better exploit flexural capacity.

During the last decades the subject became so extensive and diversified in terms of models, methods, techniques and materials applied, and so many contradictory results appeared, that a comprehensive evaluation is required. Unfortunately, one can only find very few papers attempting to provide any review on the topic. Either they focus on a specific area [9,10], or do not cover some important issues [11], e.g., modelling of knots in structural timber which is critical for understanding the effects of FRP on the wood behaviour. The authors of this particular paper intend to give a review as comprehensive as possible and identify and discuss a few unresolved topics as well.

This review study may serve as an operating manual. Researchers and engineers will find detailed information about different topics within the field of CFRP-wood strengthening and practical recommendations. The authors intend to identify and discuss a few problematic issues (e.g., the controversy over the moderate elastic stiffness increase, the modelling of knots, the strengthening of knotted timber elements, and limitations concerning modelling the CFRP-wood interface) and make general conclusions as well. Our contribution also includes a one-of-a-kind table that compares the stiffness improvement seen in various investigations with analytical estimations based on the Euler beam model.

In this paper, a short historical overview is given on the early developments first, then several different topics are discussed individually in the following sections:the CFRP-wood strengthening techniques,CFRP-wood bond characteristics and the limitation of the existing models,experimental and numerical modelling of CFRP-wood composite,effect of the CFRP on the increase of the elastic stiffness and the issue of its moderate enhancement,effect of the imperfection of the CFRP on its behaviour,strengthening timber with GFRP,modelling of knots in timber and the appeared contradictions.

## 2. Early Developments in FRP-Timber Reinforcement

Structural timber reinforcement in the 1960s relied on traditional techniques, mainly using steel or aluminium plates or rods. They are time-consuming, involve several stages, and may lead to an increase in the dead load of the structural member, in addition to their low corrosion resistance in certain cases. Ref. [12] introduced the method of bonding aluminium plate to wood section. Refs. [13,14] applied prestressed steel and strips of aluminium for reinforcing wood members in their tension zone. Furthermore, the epoxy adhesive was used by [15] for bonding steel plates to the tension face of the timber element. Ref. [16] provided a unique attempt to reinforce timber beams with glass fibre reinforced polymers (GFRP) composite. The author considered using water-based glue as well as epoxy resin. Because of its poor performance, the water-based glue was rejected. Moreover, the steel strips continued to be used for reinforcing laminated timber beams [17,18]. Also, high-strength steel wire embedded in epoxy matrix for timber reinforcing was applied [19].

The use of GFRP and epoxy adhesive appeared in research studies more frequently in the 1970s. Ref. [20] extended the use of fibreglass to reinforce plywood material. The use of GFRP also appeared in 1974 by [21] to fabricate the faces of sandwich wood beams. Moreover, wood transmission poles were strengthened with GFRP, which led to enhancing both the strength and the stiffness of the structure [22]. Another attempt for using prestressed GFRP for structural wood strengthening appeared in [23].

In the 1980s, the application of the steel bars kept to be considered in some studies [24,25]. However, [26] in his review study stated that most of the revised traditional reinforcing techniques would remain uneconomical in the future, and also stated that the fibreglass, steel fibres, carbon fibres, or aromatic polyamide fibres are some potential reinforcement materials for reinforcing timber members. This prediction, we may add in view of recent developments, has been proven completely true. To improve the tensile as well as flexural strengths of impression finger joints, fibreglass coated with phenol-resorcinol formaldehyde was utilized, increasing the strength by 10 to 40% [27]. The graphite-epoxy composite was used to strengthen the laminated wood tension members causing a recognizable enhancement in the material’s tensile strength by around 30% [28].

In 1992, the application of CFRP started to take place in timber strengthening with [29,30] relying on the analytical model introduced by [31,32] proving that the same amount of CFRP can provide a gain in ductility more than provided by the GFRP. Moreover, It has been demonstrated that reinforcing ratios of 1% of the cross-sectional area enhance resistance by up to 60%.

Note, however, the FRP reinforcing method, specifically Externally Bonded Reinforcement (EBR), was introduced first for strengthening concrete members as presented, for instance by [33,34,35]. Furthermore, most of the bond strength models used for analyzing the FRP-wood interface behaviour were initially made for concrete.

In the last three decades numerous research studies investigated various aspects of reinforcement of timber experimentally, analytically, and numerically. They are addressed in later sections of this paper.

## 3. General Introduction

The disintegration of certain pre-existing structures may be attributed to increased weights or applied loads. Replacing damaged wood elements is usually expensive and time-consuming; it may be better to repair or reinforce the timber parts instead. Steel plates or bars and aluminium plates are commonly used as traditional strengthening methods for timber structural members. Nevertheless, these approaches may increase the dead loads and have high transportation and installation costs. Some traditional rehabilitation methods and procedures may need mechanical connectors like bolts and nails. However, because the steel components are very corrosive, and the aluminium plates may bend while applying heat pressures, this may not result in good reactions from the degraded timber pieces.

When it comes to reinforcing timber, FRP composites have many significant advantages over steel, including high elastic modulus and strength, corrosion resistance, durability, easy attachment to wood, and low density. It might also be a long-term option for strengthening timber elements, particularly where load-bearing capability and ductility are required. FRP reinforcement’s high strain capacity allows wood compression fibres to yield and tension stresses to achieve ultimate tensile capacity. FRP reinforcement in the tension zone of a beam reduces tensile stresses in timber allowing compression stresses to build up resulting in a better exploitation of flexural capacity before the timber fails for tension.

Moreover, FRP composites reduce the long-term maintenance costs and can be constructed rapidly on-site. CFRP materials effectively reinforce timber structural components to enhance load capacity, stiffness, and ductility. Compared to the linear elastic brittle tensile failures encountered by the unreinforced timber beams, the CFRP reinforced beams exhibit more ductile behaviour. The type of fibre and wood used, the arrangement of the reinforcement in the element, the volume of FRP, and the quality of the bonding surface between FRP and timber influence the improved carrying capacity of reinforced beams.

FRP materials are composites made by embedding high-strength fibres (often glass or carbon) in an adhesive matrix (usually epoxy). Fibres are meaningless until they are bonded together to create a load-bearing structural element. The reinforcing components come in a variety of forms and sizes. They are often sheets, strips, or rods parallel to the timber grains placed in the tensile zone of the timber to increase tension capacity, but they may also be used for compression or wrapped around the beam. The thinnest common single plies are around 0.165 mm thick or, in some instances, much thinner [36]. Several studies, however, studied lamellae of varying thicknesses in the range of a few millimetres [37,38,39]. Rods and pultruded components can be made in increasingly more considerable sizes. The matrix may serve various functions: it can support and protect the fibres, it can transmit stresses between damaged fibres, and it is believed to have lesser density, stiffness, and strength than the fibres. As a result of the combination of both components (fibre and matrix), a competitive composite material with very high strength and stiffness but low density is produced. A matrix can be made of polymers, metals, ceramics, or carbon.

The fibres provide the strength of the composite, and the matrix is responsible for the stress transfer [40]. Studies on timber reinforcing seek to investigate experimentally, analytically, or statistically the enhancement of the structural components’ flexural capacity, stiffness, and ductility. Reinforced timber beam measurements often incorporate load-deflection relationships to experimentally validate mechanical models, whereas numerical simulations need material parameters for constitutive laws.

Material constants for numerical analysis are often derived from the manufacturer, independent tests, or published data. The manufacturer provides specifications for either the fibres or the prefabricated fibre-matrix composites in CFRP. However, in the case of wood, characteristics vary greatly depending on various parameters such as the species, age, moisture content, density, location, etc. Properties can also differ from sample to sample and are frequently just suggestive [41,42,43,44]. The elastic limit in tension and the material’s nonlinear compression behaviour are critical parameters in the overall bending behaviour. The quality of CFRP reinforcement is also affected by a variety of variables. While factory-produced lamellae are more likely to adhere to nominal requirements, in-situ hand fabrication is prone to errors. Reinforcing with CFRP generally results in a modest stiffness increase. It frequently agrees with analytical predictions (see e.g., [45]). Several research studies, however, show observed behaviour that deviates from expectations [38,46,47].

Timber numerical modelling is primarily based on the finite element method. On the one hand, however, due to wood’s anisotropic and complicated behaviour, computational models must be fine-tuned to predict the three-dimensional state of stresses. On the other hand, wood follows the elastic-plastic constitutive law in compression, and it is brittle in tension parallel to the grains. Elasticity refers to the ability of an element to deform, due to low stress, to recover to its original undeformed shape once the load is removed. Furthermore, compression plastic deformation or tensile failure will develop when subjected to a higher load value.

Tension is a typical cause of wood element failure; it results in poor utilization of the plastic ability of the beam compression zone [48]. Loading causes micro-cracks to grow into macro-cracks, causing irreversible damage inside the timber beams. Several attempts were made for the prediction of the ultimate flexural capability of timber structural elements [49,50,51]. For anisotropic materials, the Tsai-Wu strength hypothesis is the most acceptable failure criteria for determining the ultimate load-bearing of timber [49,50,51]:(1)$$F_{i}\sigma_{i}+F_{ij}\sigma_{i}\sigma_{j}=1\;\mathrm{for}\;i,j=1,2,...,6,$$
(2)$$F_{1}\sigma_{1}+F_{2}\sigma_{2}+F_{11}\sigma_{1}^{2}+2F_{12}\sigma_{1}\sigma_{2}+F_{22}\sigma_{2}^{2}+F_{66}\sigma_{6}^{2}=1,$$
where *F* stands for the strength tensors and σ values are related to the applied stresses in the three-dimensional state. When the value reaches or exceeds 1, failure occurs. Ref. [52] developed the quadratic failure criterion:(3)$${\left(\frac{\sigma_{0}}{f_{0}}\right)}^{2}+{\left(\frac{\sigma_{90}}{f_{90}}\right)}^{2}+{\left(\frac{\tau }{f_{v}}\right)}^{2}=1,$$
where σ parameters are the stresses (longitudinal and transverse directions), and *f* parameters are the strength (normal and shear).

Several authors have presented analytical models for determining the flexural capacity and stiffness of timber beams. Some of them used the models to predict the ultimate bending moment and stiffness of clear wood beams introduced by [31,32] such as [53]:(4)$$M_{u}=F_{cu}\left(\frac{bd^{2}}{6}\right)\left(\frac{3N-1}{N+1}\right),$$
where *N* is the ratio of the ultimate compressive and tensile strengths: $N=F_{tu}/F_{cu}$.

Also, the transformed sections method as introduced below was used by [54] for bending stiffness prediction, in addition to some other concepts introduced by [55,56,57]:(5)$$EI=\sum_{}^{}\left(\frac{E_{i}bh_{i}^{3}}{12}+E_{i}bh_{i}d_{i}^{2}\right),$$
where EI stands for the bending stiffness of the beam; $b_{i}$, $E_{i}$, $h_{i}$ are the width, modulus of elasticity, and thickness for each sheet of the glulam beam, respectively. The distance between the sheet’s center and the neutral axis is represented by $d_{i}$.

## 4. CFRP Strengthening Techniques

Various reinforcing techniques have been suggested, each detailing how they might improve the structural capacities of wood components. Three main reinforcing techniques are discussed in this paper: the near-surface mounted (NSM), the externally bonded reinforcement (EBR), and the glued-in rods (GiR) technique, See Figure 1.

### 4.1. Externally-Bonded Reinforcing (EBR) Strengthening Technique

Externally-bonded reinforcing is a reinforcing technique in which the CFRP laminate/fabric is attached to the element’s surface. Thin coatings of adhesive (epoxy resin) are used to guarantee good support-to-fibre uniformity during in-situ FRP strengthening. The reinforcements are exposed to the environment, sometimes resulting in inefficient use of the full tensile strength of the CFRP materials due to their fast debonding, mainly in the case of CFRP lamella. An example was introduced by [30], and different area fractions of the CFRP were applied. Results showed that the ratio of the wood tensile to compressive strains at failure, the ratio of the CFRP ultimate strain to the wood compressive yield strain, and the ratio of the modulus of elasticity of CFRP to the wood modulus are the only three parameters that affect the area fraction of CFRP. However, modest area fractions of fibre-composite reinforcement increase the mechanical performance of the member significantly.

The FRP-wood debonding failure mechanism is considered one of the main reasons for the ineffective utilization of the FRP materials. It is mostly owing to wood’s low shear strength and tension parallel to the grain. The load-bearing capacity of strengthened wood structures can be further improved by the pre-stressed reinforcement technology prior to bonding the FRP to the timber element, allowing for the most efficient use of these materials [60]. Some of the key advantages include lowering the quantity of FRP reinforcement required [61,62,63].

A method for in-situ strengthening of timber beams with pre-stressed CFRP lamella was introduced by [64]. The pre-stress forces in the CFRP were set to the maximum and zero at the middle and both ends, respectively since the delamination is expected in the region which is subjected to high stresses. According to theoretical models, prestressing using FRP composites can greatly improve the performance of low-strength wood [65].

### 4.2. The Near-Surface Mounted (NSM) Strengthening Technique

The near-surface mounted approach is an efficient method for shear and flexural strengthening and involves embedding the CFRP material into grooves in the elements to be reinforced. This method reduces the possibility of wood-CFRP delamination and increases the fire resistance of the composite system. A numerical investigation was developed to analyze the damage evaluation in timber beams reinforced with CFRP strips applying the NSM strengthening technique based on the Continuum Damage Mechanics (CDM) [58]. A similar approach was introduced by [66] to numerically replicate the behaviour of CFRP strips-strengthened timber beams resulting in a significant increase in ultimate capacity. The CFRP strips were bonded to the cuts introduced to the beams. Beams strengthened with two vertically aligned bonded-in CFRP strips resulted in a 79% and 32% increase in the load-bearing capacity and stiffness, respectively. The reinforcing technique led to a perfect bond between the wood and the CFRP, and the primary failure mode was recognized to be due to the presence of knots and slope of grains in the tension zone of the element.

Glulam beams were reinforced using NSM technique in [67]. Moreover, pull-out tests investigated the bond characteristics between glulam and CFRP strips NSM based strengthening approach. The CFRP strips were embedded in a woodblock, and different bond lengths were applied. The authors adopted two approaches, direct pull-out tests where the specimen is made up of a glulam block with a CFRP strip inserted in it and beam pull-out tests where the specimen is split into two equal-sized glulam blocks joined by a steel hinge positioned at mid-span in the upper half, as well as a CFRP laminate fastened at the bottom. The beam pull-out tests revealed better structural performance in terms of ultimate capacity and ductility since the CFRP strip is subjected to axial forces as well as a curvature caused by the rotation of the beam [68].

### 4.3. Glued-in Rods (GiR) Strengthening Technique

The Glued-in Rod reinforcing technique is based on inserting FRP rods in the timber member. It comprises a single rod or a group of rods adhesively attached to wood elements and commonly employed in wood members to carry axial loads. GiR technique is mainly used with stiff epoxy, polyurethanes, and phenol-resorcinol-based adhesives to achieve high joint stiffness [10]. The stress concentration of bonded joints is strongly dependent on the elastic properties of the adhesive and can be avoided by using a ductile adhesive with lower stiffness [69]. The use of GiR is limited due to the lack of design codes and guidelines. Previous proposals and recommendations were examined, and it is obvious that there are undesirable and recognizable variances in the computed values of the pull-out strength of single glued-in rods. However, the average shear strength value can be computed considering the geometrical characteristics (length *l* and diameter $d_{R}$) of the rod and its type ($k_{b}$), moisture ($k_{e}$), and the used adhesive ($k_{m}$) as [70]:(6)$$F_{R}=6.73\cdot k_{b}\cdot k_{e}\cdot k_{m}\cdot {\left(\frac{l}{d_{R}}\right)}^{0.86}{\left(\frac{d_{H}}{d_{R}}\right)}^{0.5}{\left(\frac{e}{d_{R}}\right)}^{0.5}.$$

Another formula for predicting timber elements and joints capacity based on a probabilistic approach was also provided by [71]:(7)$$P_{s}\left(\sigma \right)=e^{-{\left(\frac{\sigma_{i}}{k}\right)}^{m}},$$
where $P_{s}$ stands for the probability of success. *m* and *k* are related to the material strength.

Moreover, a study was performed to predict the ultimate capacity of timber beams strengthened with GFRP glued-in-rods based on a probabilistic method, and also to evaluate the effects of the following three components: adhesive, rods, and timber, along with numerical investigation for the stress distribution inside the joints. A ductile adhesive was utilized, undergoing a large plastic deformation without affecting the overall behaviour of the GiR. However, this plastic behaviour was revealed numerically to have a significant effect on the stress distribution within the joints [72]. Another similar work applied various types of wood manufacturing defects was done to determine their impacts (capacity and stiffness) represented by the imperfection, the glue, and the wood product type on threaded rods bonded into beech and spruce laminated timber components using four different adhesives. There was no critical reduction in stiffness and ultimate capacity due to the induced artificial imperfections. However, the voids were found to be significantly affecting the behaviour of the materials, and the hardwood showed a better performance [59]. Another study evaluated the efficacy of strengthening passively reinforced and prestressed beams. Up to 131% and 42% in flexural capacity and bending stiffness were reported changing the failure mechanism from brittle tension failure to ductile compression failure [73].

### 4.4. Comparison between the Failure Modes

Following is a brief comparison of EBR and NSM strengthening techniques. The EBR technique is the most common method to strengthen timber structures. Installation is quick and easy with low costs and does not require skilled labourers. However, this reinforcing technique has some disadvantages. It may exhibit a brittle debonding failure between the CFRP and the timber member. The material is directly subjected to external environmental effects that may negatively affect the overall behaviour of the composite system. The NSM method does not require a surface preparation reducing the strengthening processing time, and it results in a perfect bond between the composite members minimizing the debonding issue. Moreover, the material is protected against environmental effects, and several FRP material types can be used, such as rods and strips. Nevertheless, this method is costly and requires a special and careful installation process.

## 5. CFRP-Wood Bond Characteristics

CFRP-to-wood bond behaviour is critical and has a significant effect on the overall structural behaviour of the CFRP-wood composite system. Different failure modes can be observed in CFRP reinforced timber beams, such as FRP/adhesive separation, FRP rupture, and adhesive failure. However, premature failure due to debonding is recognized as the most known failure mechanism in FRP beam strengthening. Debonding occurs when the tensile stress at the interface exceeds the bond strength, and it prevents the strain from being transmitted from the wood to the composite fibres, which prevents the full utilization of the FRP strength; also, the structural member will not attain the desired ductility [74]. Different parameters can affect the bond strength such as the characteristics of the wood [75,76,77], moisture content [78], prestressing [56], surface preparation [79], the adhesive and temperature [80,81], the bond and the FRP geometrical and material properties [82], the boundary conditions [83], the bond length (the effective bond length must be taken into consideration) [84,85], the proportion of FRP to substrate width [86].

The adhesive’s performance directly impacts the composite’s performance and the bond strength. It plays an essential role in obtaining a sound CFRP bonded wood system to effectively transfer the shear, and regular forces from the timber to the composite material [87,88]. Nevertheless, the failure of the timber element was recognized to be the dominant failure mode in another study regardless of the properties of the adhesive material, implying that the qualities of the wood, rather than the adhesive, may control the bonding behaviour [89,90]. Timber joints with varying bond widths, bond thickness, lengths, and cross-sectional dimensions were reinforced using CFRP to predict the FRP-to-timber interface’s bond strength and ascertain the key components influencing bond strength with the help of a proposed regression bond strength model. The wood species, bond geometrical characteristics were analyzed against their effectiveness on the bond strength. Five bond lengths were tested under shear to evaluate the bond behaviour and bond-slip interactions precisely. However, the out-of-plane movement of the specimens could not ensure a proper pure shear test set-up resulting in misleading LVDT findings. The bond strength was proportional to the FRP plate width, and the local slip at the same load level reduces as the FRP-to-timber width ratio increases. It was stated that the bond strength was enhanced as a result of the hardwood species’ higher tensile strength [85].

A few models for analyzing CFRP-wood bond behaviour are available in the literature. Bond stress-slip and bond strength models were developed by Wan [90] as:(8)$$P_{u}=0.012\gamma_{t}\gamma_{e}b_{f}L_{e}^{0.28}\sqrt{E_{f}t_{f}}\;,$$
(9)$$\tau_{x}=\frac{dj}{dx}=A^{2}BC_{N}e^{-Bs}\left(1-e^{-Bs}\right).$$

Originally, these models were proposed for the FRP-concrete interface. However, concrete is weak under tension, whereas wood is generally stronger, resulting in different failure mechanisms, limiting the model’s use and opening the field for developing new models. The terms *t* and *e* in Equation (Equation 8) stood for timber and adhesive type, respectively. Bond with, effective bond length, and stiffness of FRP were referred to as $b_{f}$, $L_{e}$, and $E_{f}t_{f}$, respectively. *A* and *B* were experimental parameters in Equation (Equation 9), *s* was the equivalent slip at a given position, and $C_{n}$ was the elastic stiffness. More bond strength models of FRP laminates to wood were proposed by [91] as:(10)$$P_{u}=\frac{1}{2}b_{f}f_{ts}\sqrt{\frac{E_{f}t_{f}}{2f_{ts}}},$$
and by [92] as:(11)$$P_{u}=P_{max}\frac{L_{b}}{L_{e}}\left(2-\frac{L_{b}}{L_{e}}\right),$$
(12)$$P_{max}=c_{1}k_{b}k_{c}k_{\mu }b_{f}\sqrt{E_{f}t_{f}\tau_{max}},$$
(13)$$L_{e}=\sqrt{\frac{E_{f}t_{f}}{c_{2}\tau_{max}}},$$
(14)$$k_{b}=1.06\sqrt{\frac{2-\frac{b_{f}}{b_{w}}}{1+\frac{b_{f}}{400}}}\leq k_{b}\leq 1.29,$$
where τ is the bond stress; $\tau_{max}$ is the bond length’s maximum shear stress; $P_{u}$ is the maximum load; the width, elastic modulus, and thickness of a FRP plate or sheet are denoted as $b_{f}$, $E_{f}$, and $t_{f}$, respectively; $b_{w}$ is the width of the wood block; $f_{ts}$ represents the tensile strength of the wood; the bond length and effective length, respectively, are $L_{b}$ and $L_{e}$; $c_{1}$ and $c_{2}$ are parameters calibrated from the experiments; the anchor zone geometry is represented by $k_{b}$, which has a range of 1–1.29; the factor $k_{c}$, with a range of 0.67 to 1, represents the surface preparation effect; the component $k_{\mu }$ denotes the degree of strengthening.

The prestressed reinforcing method can effectively prevent early failure caused to debonding. The surface treatment procedure may be crucial in improving the FRP-wood interface bonding performance. For example, sanding and cleaning the wood and FRP surfaces, as well as high-performance adhesive modification treatments.

## 6. Numerical and Experimental Studies on FRP-Wood Strengthening

### 6.1. Strengthening Timber with CFRP

The application of the GFRP-wood strengthening first appeared in the 1960s with [16,93]. At the beginning of the 1990s, the usage of the CFRP has started with [29], who proposed the novel technique of external wood reinforcement by externally bonded prestressed unidirectional carbon/epoxy CFRP sheets. They extended the analytical model proposed by [94] defining the ultimate prestress level that may be achieved by a concrete beam avoiding the failure of the FRP-prestressed system near the two ends upon releasing the prestress. A prestress $\sigma_{p}\left(x\right)$ appears at the extreme tension fibre of the beam after releasing the initial intensity of pretension. The surface of the wood was cleaned, and sandpaper was used to roughen the surface of the CFRP. A proper selection was recommended for the adhesive to ensure a perfect bond between wood and FRP; the FRP area fraction and the initial pretension intensity were also specified. The FRP area fraction was supposed to be related to the pretension level for obtaining configurations suitable for better flexural enhancement.

Wood anisotropy is comparable to transversely isotropic fibre-matrix composite material anisotropy in terms of stress-strain characteristics, failure modes, and preferred fibre orientation, but it is more complicated. Fortunately, the linear orthotropic material model with nine material constants can always be used to describe the elastic behaviour of timber. Several previous research studies have addressed the analysis of the complicated behaviour of carbon-reinforced wood beams by generating linear and nonlinear mechanical models based on experimental work and the finite element method (FEM), which provides a robust numerical approach [95,96,97,98,99]. Numerical analysis demands the use of constitutive models with proper material parameters.

Fibre-reinforced composites are often linearly orthotropic elastic with brittle rupture, but timber requires a more complex definition. Unfortunately, the linear orthotropic model is only useful in the linear range. Non-linear stress-strain relationships are needed beyond that, particularly in the compression region where numerous other variants, such as completely plastic, have been utilized (e.g., [38,66]), bilinear (e.g., [11,54,100]), higher-order (e.g., [101]). Ref. [102] presented a tri-linear stress-strain diagram to analyze the failure behaviour of wood under compression parallel to the grain, see Figure 2. Furthermore, a bilinear anisotropic stress-strain relationship was proposed by Hill that may be used to predict the orthotropic linear elastic–quasi-rigid behaviour in tension and orthotropic linear elastic–perfectly plastic or bilinear behaviour in compression [103,104,105,106].

Several experimental attempts were made to investigate the influence of CFRP on the flexural behaviour of wood structural components. Nevertheless, experimental testing mostly comes at a high cost. Numerical modelling is a powerful technique for analyzing the behaviour of timber structural parts and FRP reinforced wood beams. However, there is a lack of research on the formulation of finite element models that include nonlinear material modelling of timber. The orthotropic constitutive properties of wood must be carefully modified in the numerical simulation to accurately simulate the timber structural member’s complicated behaviour.

The reinforcing materials are typically attached to the element’s tension side because they bear a part of the tensile stresses applied to the timber-CFRP structural elements. Less often, sides of the cross-section can be partly or completely covered by the reinforcement, or fully wrapped around (Figure 3a). Ref. [108] suggested a technique for analysing the impact of CFRP sheets on historic wood structural components’ flexural stiffness and capacity. It was stated that better structural enhancements could be obtained by ensuring a good bonding between the wood and the CFRP without removing the deteriorated part of the timber beams. They also proposed an equation to predict the ultimate capacity of the strengthened beams. A maximum of a 60% increase in capacity and a 30% increase in stiffness were recorded. Different wood Young’s modulus to CFRP modulus ratios has been observed due to plasticization in the compression zone. Ref. [109] investigated three methods for reinforcing an existing Spruce timber floor: a piece of wood fixed to the upper side of the beams with dowels (dry and glued) made of hardwood, a CFRP sheet on the bottom. They reported that the bond at the wood-fibre contact significantly impacts mechanical behaviour. Results obtained from the pull-out test showed that the combined shear-flexural was the dominant failure mode when a space is left between the original wood and the reinforcing wood pieces and the glued dowels showed a better performance. A perfect bond was ensured for the CFRP-timber, which practically obtained a good strengthening technique with a recognizable increase in ultimate capacity and stiffness. Both rupture of CFRP and wood fibres in the tension zone were revealed as main failure modes. The combined system allowed the detection of new mechanisms of failure (sudden delamination of the CFRP sheet) and was reported as the most significant capacity increase of 3.5 times.

Three-point or four-point bending tests are typically applied to experimentally acquire load-deflection data on reinforced beams (Figure 3b), which can be compared with analytical or numerical investigations using classical beam theory or finite element models (Figure 3c,d, etc.) [110] presented analytical, three-point bending experimental, and FE studies investigating the effect of unidirectional CFRP laminate of two different lengths on the behaviour of pine wood. Compression and tension tests were performed for both wood and CFRP to determine the stress-strain diagrams for the specimens. It was shown that the reinforcements with shorter lengths do not contribute to the increase of load-bearing capacity and do not improve the ductility of the wood-CFRP system. Moreover, reinforcements with shorter length yield a 23% increase in stiffness and 12% in ultimate capacity compared to the increase of 34% and 28% in stiffness and load-bearing capacity obtained for the longer ones. This might be related to the failure mode explained by a failure initiation at the CFRP boundaries causing failure in the adhesive, which propagates to the wood resulting in final collapse.

To describe the behaviour of wood, numerical attempts were made using various material models. A bi-linear approximation was experimentally derived to describe the stress-strain behaviour of timber in compression and a linear relationship up to maximum stress in tension to study the effect of reinforcing type, position, and amount of CFRP in the form of strips on the strength and stiffness of timber beams in both elastic and inelastic states. An increase of 40% in stiffness was reported for the Sitka spruce beams reinforced in tension with a 0.8% volume ratio of CFRP. Results also proved that using the same amount of reinforcement in tension causes strength and stiffness increase, similar to the case of distributing them in tension and compression. A comparison indicated that CFRP could help to achieve three times higher load-bearing capacity than steel [46]. A similar attempt was introduced to evaluate the bending strength and stiffness of reinforced glulam beams using externally-bonded FRP reinforcing technique and theoretical models [54]. A linear-elastic-ideal-plastic model with constant stiffness and strength properties was considered by [55]. The load-deflection curves for glulam beams made of Sitka spruce reinforced with CFRP plates were numerically evaluated with the help of a two-dimensional plane stress model taken into consideration both material and geometric nonlinearities [107]. An increase of 8.4% in stiffness was numerically obtained for the reinforced beams. The model can incorporate yield stresses in compression, and it is known as the generalized anisotropic Hill potential [103]. An elastoplastic constitutive law for orthotropic materials was introduced by [104] where the plastic part of the stress-strain curve is considered as two straight lines with different slopes, see Figure 2 for model illustrations. More suggested models can also be found in [31,111]. The linear elastic orthotropic material model is used to evaluate the behaviour of CFRP composites’ behaviour until failure [112].

Moreover, Ref. [113] provides the relationship between the tensile stress $\sigma_{t}$ and the ultimate tensile stress at failure $\sigma_{tu}$ for describing the stress distribution in the cross-section:(15)$$\sigma_{tu}={\left(\frac{C}{1+K}\right)}^{-\frac{1}{K}}\cdot \sigma_{t},$$
where *C* is a parameter related to the ratio of the tension zone depth and *K* indicates the variation in strength characteristics within a member’s depth.

Ref. [32] also suggested a stress-strain curve for timber, based on elastic-plastic compression and linear elastic tension. More research studies on determining material characteristics have been suggested. Ref. [114] proposed a polynomial function to express the nonlinear stress-strain relationship beyond the elastic in the compression zone. The function is as follows: (16)$$\sigma_{cw}=A\cdot \epsilon_{cw}^{2}+B\cdot \epsilon_{cw}+C\;\mathrm{for}\;\epsilon_{cw0}\leq \epsilon_{cw}<\epsilon_{cwu}.$$
where $\sigma_{cw}$ and $\epsilon_{cw}$ are the compression stress and strain; $\epsilon_{cw0}$ and $\epsilon_{cwu}$ are the strain limits in compression. The coefficients *A*, *B*, and *C* can be determined from the experiments.

A study was given to evaluate the changes of the timber member elastic characteristics in the planes of LT and LR, represented as a material that is linearly elastic and orthotropic as a function of the Euler’s angles (λ, ρ) proposed by Hermanson, as well as several tests [115,116]. The Generalized anisotropic Hill model was generated for analyzing the behaviour of the spruce (*Picea abies*) timber beams utilizing the finite element method. The model was validated, and the properties of the examined material were established using three-point static bending and compression tests [117]. When the Laminated Veneer Lumber (LVL) is homogeneous, and the CFRP fibres are unidirectional, a hypothetical model that was utilized as an example to evaluate (LVL) Young’s Modulus reinforced by CFRP was also presented by [118]. Another author gave a similar example [119]. To investigate the behaviour of Norway spruce wood components, scholars also suggested finite element models [120,121,122].

Ref. [37] experimentally investigated the effects of CFRP and GFRP on the behaviour of *Pinus caribaea* var. *hondurensis* wood species. Results were compared to a theoretical model considering a fragile tension and elastic-plastic compression. The FRP sheets were applied manually on site. As expected, the dominant failure mode was the tension failure in timber. The vertical displacements were determined from the bending stiffness based on the transformed section method and the failure moment based on Navier/Bernoulli beam model. The experimental values of the bending stiffness and the failure moment were higher than the theoretical ones. The reinforcements led to a significant increase in ductility in the FRP-wood system. The results revealed that stiffness increased by 6% to 15% depending on the type and amount of FRP used.

Furthermore, for the sake of comparison and verification, the CFRP elastic characteristics were predicted using a mathematical technique in connection with an experimental study [123]. Analytical methods were used to predict the unidirectional compression strength of the CFRP [124]. Ref. [125] proposed equations that were used to determine the modulus of elasticity of wood fiberboard. The CFRP material requires nine parameters for its complete characterization. The behaviour is mainly affected by the modulus of elasticity in the fibre direction. However, the inverse rule of mixtures is used to compute the transverse modulus of elasticity as:(17)$$E_{2}=\frac{E_{f}E_{m}}{E_{f}V_{m}+E_{m}V_{f}},$$
where $V_{f}$, $E_{f}$ and $V_{m}$, $E_{f}$ are the volume fractions and moduli of elasticity for the fiber and matrix respectively. The shear modulus $G_{12}$ and the Poison’s ratios $\nu_{12}$, $\nu_{21}$ are approximated with the following equations:(18)$$G_{12}=\frac{G_{f}G_{m}}{G_{f}V_{m}+G_{m}V_{f}},$$
(19)$$\nu_{12}=\nu_{f}V_{F}+\nu_{m}V_{m},$$
(20)$$\nu_{21}=\frac{E_{2}}{E_{1}}\cdot \nu_{12}.$$

Ref. [126] applied a non-deterministic design with the help of the Monte Carlo design method to investigate the uncertainty in the unreinforced and CFRP reinforced timber material strength, and it was stated that the increase of the safety factor which represents the presence of the uncertainties does not yield a reliable design. When strengthening a wood beam with CFRP, two failure mechanisms may be identified: the first is the tension failure, which is the most common, and when the compression stress limit is achieved, the second occurs. Ref. [127] analyzed the effect of the nonlinearity (bilinear) on the deflection and stress numerically in steel cantilever beam, and it was found that the deflection and load in the nonlinear analysis are proportional and the stress—load are inversely proportional.

Strengthening defected timber beams with CFRP composite need to be addressed more in literature. Few authors suggested reinforcing the defected wood element in the vicinity of the knot. Knots usually produce stress concentrations only in their vacuity. As a result, reinforcing the entire beam is a cost-effective procedure. D-shape CFRP strips were used to locally strengthen defective spruce wood beams using experimental and finite element modelling methodologies. The strips were optimized and applied in the tension zone of the beam around the knots (modelled as a series of holes of various sizes), assuming that CFRP has a linear elastic response and wood was subjected to bending and evaluated in the elastic range. The beams did not show any significant plastic deformation until failure. The general purpose was to introduce a local reinforcement technique to reduce the effects of knots in terms of elastic properties and bending strength. A groove was manually made in the same non-reinforced tested beams before inserting the D-shape CFRP reinforcements. Then CFRP was bonded to wood using epoxy adhesive, and four bending tests were again conducted until destruction. Considering a CFRP Young’s modulus with at least 16 times that of Spruce wood, it was found that using local reinforcement to improve bending stiffness would be desirable [120].

Another experimental and analytical attempt for local reinforcement for old wood beams using CFRP was provided by [128]. The defects were in the form of a knot located at the tension side of the beam and an artificially induced imperfection. The results show that it is feasible to partially repair the bending ability of damaged wood beams. The local retrofitting technique restored up to 11.5% of the timber beams’ bending capacity. The maximum bending load in old timber beams with inherent defects was studied experimentally and analytically for comparison. Beams were reinforced in their tension side with CFRP laminates. The comparison between the experimental results and those obtained from the analytical models suggested by [32,113] showed a weak correlation (average error of 10.2%) in terms of the modulus of rupture MOR. Nevertheless, results obtained by [129] considering different modulus of elasticity in tension and compression revealed a good correlation. However, MOR improvements can range from 14% to 88% depending on the amount and location of defects [130].

Table 1 presents a qualitative and quantitative summary of some key experimental and numerical studies. It shows the importance of the pre-stressing technology (passive or slack reinforcement) which prevents the FRP delamination failure.

### 6.2. Strengthening Timber with GFRP

GFRP composite materials are frequently mixed with polymeric matrices (polyester, epoxy, vinyl ester, thermosetting plastic, etc.). The obtained materials provide engineers with exceptional characteristics, including low weight, very high strength, and good chemical resistance. Unidirectional composites have a poor transverse compressive strength due to their anisotropic nature. They are also fragile, which makes them susceptible to stress concentrations [131,132]. The strength characteristics of the glass fibres may be adjusted to produce a high-performance composite product. The climate, coating, and treatment are important factors in the variance between theoretical and actual values of glass fibre strength. Heat treatment is another factor that contributes to the weakening of glass fibres. Generally, the GFRP mechanical characteristics in terms of stiffness and tensile strength are 3 times less than CFRP properties. The carbon structure is lighter than the glassfiber structure [38,133]. GFRP for timber reinforcing purposes can appear in a form of laminates (e.g., [134,135,136]) or bars (e.g., [137,138,139]).

According to literature, the glass fibres were the only reinforcing agent under consideration between 1957 and 1965 [140]. However, recent research studies have revealed that depending on the strengthening method, it is feasible to improve timber elements’ flexural, shear, or both characteristics. GFRP can be applied directly in situ for strengthening an existing structure or prefabricated, especially within a glued laminated wood, mainly on the tension side to prevent the fragile tension failure and enhance the flexural capacity of the timber [141]. A study was conducted to compare the load-deflection behaviour, stiffness and ultimate flexural capacity increase, failure mechanism, and ductility of the reinforced and unstrengthened laminated wood beam with GFRP and CFRP composite sheets applied at the bottom surface manually in-situ. Tension was reported as the dominant failure mode. The flexural rigidity (EI), Modulus of rupture (MOR), and beam’s ductility (the ability to withstand inelastic deformation before failure) of a timber beam reinforced with a single layer of GFRP increased by 26.29%, 36.91%, and 44.37% respectively [38]. A theoretical nonlinear model based on the classical beam theory was proposed considering an exponential function for the stress-strain diagram of wood plastic composite beam reinforced with CFRP and GFRP composite sheets glued to the tension face. The model was assigned to longitudinal fibres and validated by load-displacement relationships obtained from four-point bending tests to examine their flexural behaviour. A recognizable increase in capacity and stiffness was recorded. However, tension failure accompanied by GFRP rapture and debonding were reported during the experiments [133]. An attempt was presented through an experimental technique by [142] to repair the centre and ends of timber beams with epoxy resin-bonded fibreglass and cork plates. The reinforcing materials were used to enhance the shear and bending strength of the beams. The grooves that were inserted before reinforcing resulted in premature failure and loss in the fracture load. Moreover, the GFRP sheets were introduced to strengthen pre-existing timber beams. Holes were drilled in both the GFRP and the wood to join them mechanically. The techniques led to an increase of 75% and 49.7% in flexural capacity and stiffness, respectively. However, the stress transfer was found to be somehow independent of the number of connectors, and configuration types [143]. Another study concerning the flexural retrofitting of old timber elements in historical buildings using GFRP was performed by [135,144].

The GFRP can also be applied for strengthening glued laminated timber beams [145,146]. A study showed that it is not advantageous to employ a number of smaller diameter GFRP rods to increase the contact GFRP-wood surface. The spatial configurations of the drilled grooves are significant. It was also stated that by decreasing the effects of stress concentrations, improvements in the mechanical performance of reinforced beams could be made [147]. The increase in stiffness and flexural capacity of strengthened laminated timber beams was found to be more significant when using less percentage of CFRP in comparison with the GFRP [38]. Epoxy adhesives are still the most popular choice for bonding fibre-reinforced plastics [148]. The effectiveness and durability of the GFRP-wood bond are crucial, and it depends on both the adhesive and FRP types applied for glulam wood strengthening; as opposed to wood, FRP materials absorb moisture at a much lower rate [149].

Table 2 shows a quick comparison between GFRP and CFRP composite materials:

Note that there are examples of combinations of fibres. In a recent study, a suggestion was presented for using hybrid fibre reinforced polymer (HFRP) to strengthen wood beams [53]. In comparison to the previous use of single-type fibres for strengthening, the application of hybrid carbon fibre/high-strength glass fibre strengthening can significantly improve ductility.

### 6.3. Effect of the CFRP on Flexural Stiffness Enhancement of Timber

Wood reinforcing technology aims to enhance the flexural capacity, ductility, and, in some cases, stiffness of wood components. Fortunately, it has produced positive outcomes in a variety of innovative applications. The load-bearing capability has increased by 20–60%, according to the findings [108], and in some cases even more. The kind of reinforcement and wood species, among other criteria, are the most critical aspects that significantly impact the possibility for structural behaviour enhancement. The experimental study done by [150] employed sawn Norway spruce beams strengthened with various amounts of CFRP. In general, approximately a 30% increase in the load-bearing capacity and almost 16% increase in the elastic stiffness were found. Theoretical and experimental investigations were carried out by [112] on two different wood species retrofitted with different quantities of CFRP sheets to evaluate their effects on flexural behaviour. A minimum increase of 39% of load-bearing capacity was reported. The force-displacement relationship of the wood beam reinforced with CFRP sheet was calculated with the help of the classical beam theory by using moment-curvature relationships considering Hook’s law for the linear stress-strain relationship and the second-order polynomial function for the nonlinear part. The shear effect and the CFRP-wood debonding was not taken into consideration. CFRP rupture and flexural failures were reported. Furthermore, the application of the GFRP bars leads to an average increase in 32% flexural capacity. Moreover, the study showed that the failure mechanism was switched to compression failure instead of brittle tension [137].

In contrast, it was also demonstrated that, in most situations, the addition of reinforcements had a slight effect on stiffness [134,150,151,152,153,154], and the increase does not usually exceed 30%; it is primarily negligible and occasionally much more significant. Bidirectional carbon fabrics and laminate strips were applied for strengthening timber beams taken from an old bridge. A noticeable increase in shear and bending capacities was found; however, a slight increase in stiffness was recorded [155]. A similar study was conducted to reinforce timber stringers in either or both flexural and shear using GFRP dowel bars reported an increase of the ultimate capacity (70%) and ductility with a low increase in elastic stiffness [151]. Two different types of U-shaped reinforcements (unidirectional and bidirectional CFRP fabrics, basalt fibres) were applied for strengthening wood beams in bending. No fibre delamination occurred, and the tension failure was the dominant failure mode. Results also showed that the beams reinforced with two layers of bidirectional CFRP fabric have an ultimate load identical to that of beams reinforced with a single layer. The reinforcements of basalt fibres showed better performance than the CFRP materials.

Furthermore, the increase of stiffness was found to be proportional to the amount of used fibres [152]. In most situations, the strengthened beams’ ductility is improved, see, e.g., [156]. According to some authors, stiffness can rise by 17–27% and strength by 40–53% [155]. In-situ reinforcing techniques employing FRP lamella for repairing and strengthening of existing wood elements were studied, with a 6% improvement in bending stiffness and a 25% increase in flexural strength [157]. A similar study also resulted in a 58% increase in load capacity while a moderate enhancement in beam stiffness [158]. As a result of demonstrating that CFRP to timber is a feasible approach to strengthening, analytical and numerical modelling efforts are warranted.

Experimental and computational investigations on wood-CFRP strengthening were carried out. When compared to findings obtained for beams without reinforcing with CFRP fibres that had an elasticity modulus of 59,000 MPa and tensile strength of 1034 MPa, the study found a 21.5% gain in strength and a minor improvement of 4.69% in stiffness [159]. The influence of externally bonded CFRP lamellas on the stiffness and capacity increase of timber beams was introduced in another study; the stiffness increase was not recognized, and a comparison of the theoretical to the experimental wood-to-CFRP modulus of elasticity ratio revealed a significant variability [47]. A 10% increase in stiffness was reported, as well as a 44–60% increase in capacity [100]. The results demonstrate that introducing bidirectional carbon fabric to timber beams resulted in a considerable improvement in bending shear capacity and a minimal increase in beam stiffness. The use of CFRP sheets for wood strengthening resulted in a stiffness increase of 15 to 60% [109], and an increase by 20% in another study [160].

The addition of carbon plates and sheets increases stiffness ranging from 20.2% to 29.6% [161]. In addition, an experimental study of reinforcing laminated wood beams using CFRP and GFRP composite sheets was conducted. The flexural stiffness of a CFRP composite sheet increased by 36.19% and 64.12%, depending on the amount of fibres employed (1.67% and 3.33% respectively). Flexural strength increased by 45.86% and 50.62%, respectively [38]. The behaviour of wood beams reinforced with CFRP composites was predicted using numerical techniques based on the finite element method. The results showed an increase in stiffness for specimens with low modulus of elasticity for beams strengthened with CFRP sheets and strips [39]. The stiffness increases with FRP strengthening of timber beams was modest, and it was also heavily dependent on the span to depth ratio of the beams; reinforcing timber beams with GFRP laminates increased the stiffness of the beams by 3% [134]. A study on the influence of GFRP composite plates on wood behaviour also revealed a moderate stiffness increase (13.15%) [45]. A nonlinear numerical model used to investigate the influence of FRP plates on the strength and stiffness increase of timber indicated an 8.4% stiffness increase [107]. The reinforcing effectiveness of CFRP on timber capacity and strength uncertainty was demonstrated through 221 bending tests on unreinforced and reinforced soft- and hardwood beams [36]. The effectiveness of utilizing CFRP and GFRP to strengthen timber beams/stringers and piles used in structural bridge components was examined, as well as the influence of FRP wrap on the beam stiffness [162].

The authors believe that one of the main reasons for moderate stiffness enhancement is attributed to the CFRP fabric’s preparation. The lamellae or fabric were made in-situ by rolling glue over the surface, producing the embedding matrix and connecting it to the wood simultaneously. The method is effective, may be used for reinforcing in any location, and ensures a good connection between the fibres and the wood. However, despite all of its benefits, it has disadvantages. Because the extremely thin carbon fibres are sensitive to any action other than axial tension (e.g., bending), in-situ fabrication of the reinforcing lamellae/fabric invariably causes some damage. Geometric flaws in the fibres (such as waviness) are unavoidable, decreasing the elastic modulus further. Because the fabric is thicker than normal and the epoxy has high consistency, it takes much power to get the epoxy to permeate the fabric; thus, its effect is significant. Table 3 shows the obtained increase of stiffness in some studies versus the expected percentage with the help of the classical beam theory.

As can be observed, the stiffness findings varied in each example, primarily attributable to different reinforcing methods and the properties of the timber employed. The adhesive ingredient, commonly a polymeric resin, does not contribute much to improving strength characteristics; nevertheless, it keeps the entire system together, allowing it to function as a composite. The elastic moduli and allowable stresses of the component materials affect the increase of stiffness.

## 7. Effect of the Waviness of the CFRP on Their Performance

Fibre waviness is a process or material design induced imperfection that results from the distortion of fibres that are meant to be straight in the laminate. Defects in the laminate might occur during the CFRP production process and because of the difference in thermal expansion coefficients between fibres and work material [163]. During the production process, compressive thermal residual stresses cause fibre waviness [164]. These defect, e.g., ply waviness (see Figure 4), leads to a reduction in compressive strength and stiffness. It is thought to have the greatest effect on compression capacity. RTM (Resin Transfer Molding) is one of the most critical manufacturing processes; this technology has been utilized extensively in the aerospace and automotive sectors owing to its enormous capacity to make composite components with complicated geometries at high production rates. In terms of production speed, the manufacturing method for FRP composites has evolved [165].

Out-of-plane fibre waviness is usually found in thick laminates. However, some techniques were applied to detect the most common in-plane fibre waviness: eddy current method [166,167,168], X-ray computed tomography (CT) [169], Ultrasonic testing methods [170]. The artificial in-plane fibre waviness in cross-ply (laminates that are uniquely composed of laminae with fibre orientations of 0° and 90°) CFRP thin laminates were detected with the help of a non-destructive eddy current testing method [166]. The eddy current visible around the tube creates its magnetic field, which opposes the coil’s magnetic field, resulting in poor self-induction and high current consumption. When a fault (such as a crack) comes in the way of the eddy current, the coil’s current consumption decreases and the eddy current must flow around the imperfections. As a consequence, low current consumption when the tube is far away from the coil, high current consumption when the tube is inside the coil, and then the current is reduced and defects are recognized when the tube reaches a fracture.

A study that focuses on the effect of ply waviness on the stiffness and strength reductions of the composite laminates was presented by developing an analytical model based on the laminated plate theory [171]. The fibre waviness was evaluated in two cases: fibre waviness in a single ply fibre/matrix level and the waviness of fibres in plies inside a multidirectional laminate, assuming that fibres have a sinusoidal shape. The material properties and stresses were predicted in terms of ply waviness and material properties of the laminate. The extreme case revealed a significant reduction in stiffness of 60% and 65% in strength. Increasing undulation amplitude and decreasing unit cell length both increase observed stiffness reduction. Moreover, the effects of the fibre waviness on the modulus of elasticity, shear modulus, and poison ratio of DFR (discontinuous fibre reinforced) were investigated with the help of RVE models. The influences of fibre volume fraction $V_{f}$ and the fibre/matrix interphase area were also studied, and micromechanical finite element models were presented. Three-dimensional RVE were created by varying waviness ratios A/λ (wave-amplitude to length), wavelength ratio λ/d (length to diameter) and $V_{f}$ [172]. The waved fibres were assumed to have a sinusoidal form as: $y=A\sin\left(\frac{2\pi x}{\lambda }\right)$. It was found that a considerable effect on the effective properties of the composite with waved fibres is attributed to the fibre/matrix interphase.

## 8. Modelling of Knots in Structural Timber

### 8.1. General Overview

Knots and associated grain deviations are the most common natural imperfections, and they diminish the ultimate capacity of timber components [173]. Knots decrease tensile strength due to grain deviation and stress redistribution, although it has a minimal impact on compression strength parallel to the undisturbed grain direction. We should distinguish between two types of knots: the term “dead knot” refers to a knot that is no longer attached to the main body of wood, whereas “live knots” are still connected to the main body of the wood. Moreover, undergrown knots are more likely to produce grain deviation and tension reduction than dead knots.

The global and local fibre deviations significantly impact the material characteristics of wood, such as strength and stiffness. They also result in a localized stress concentration, which leads to an early tensile failure and inefficient utilization of compression capabilities [174]. The impact of such deficiencies on the strength grading of timber structural elements would be more efficient if the influence of such defects were fully understood. The most concerning characteristics are the longitudinal tensile strength, MOR, compression strength parallel to the grain, and the MOE. Knots frequently promote stress discontinuities in their vicinity and a deviation of the surrounding fibres in the tangential plane of the material (LT). The inhomogeneity between the knot and the surrounding wood material will also contribute to the strength loss. Furthermore, the density of the knotted wood is twice higher than the average density of the unaffected wood.

To measure the fibre deviations, several non-destructive techniques can be applied, such as the laser confocal microscope, [175], the application of the tracheid effect scanning [176], X-ray scanning technique [177], the CT-direction process (Computed Tomography) [178]. Mathematical models were also developed to describe branch, and knot formation concepts [179,180,181].

An X-ray scanner was employed to create a 2D in-plane pattern for the knots, grain angles, and density variations. However, the out-of-plane component (diving angles) was not measured. It was found that knots spaced at least 200 mm apart can significantly reduce the stiffness more than an individual knot. To estimate the tensile strength, 44 specimens of timber were examined in tension, the pith was found in 44% of the observed failed sections, whereas the knots in the middle of the wood section were found in 10% of the failure sections [177]. A schematic diagram of the vicinity of a knot is shown in Figure 5. A photo taken after the failure of a test specimen reveals the complex stress distribution around the knot.

Ref. [176] proposed a non-destructive method to develop a 3D fibre direction pattern for the Norway spruce timber by investigating the relationship between the diving angle and the grayscale image of the reflected light applying the tracheid effect scanning to determine the in-plane and out-of-plane angle (fibre direction) especially in the vicinity of the knot and verified it by an already defined mathematical model created by [181].

The behaviour of Scots pine *Pinus sylvestris* wood was studied using 3D finite element models that were considered to be transversally isotropic (five different material characteristics for the elastic behaviour). The authors presented a 3D analytical wood material model to simulate the behaviour of timber beams considering the effects of any existing knot in the form of an ellipse, rotated and oblique cones. The discontinuity of the stress distribution caused by the knots is the reason for strength reduction. Shear and perpendicular to grain stress components arise in the fibres surrounding the knots. These stresses are more critical than parallel to fibre stresses. For tension and compression, different Young’s moduli ($E_{c}$ and $E_{t}$) were considered, and the bending modulus of elasticity was obtained using the equation:(21)$$\mathrm{MOE}=\frac{4E_{t}E_{c}}{{\left(\sqrt{E_{c}}+\sqrt{E_{t}}\right)}^{2}}.$$

In another study, the grain deviation around the knot was calculated using a 3D flow-grain analogy, which included making the knot’s contour a solid obstacle while conducting laminar flow through the beam and measuring the velocity components in each element [182]. The 3D flow-grain analogy is a strategy where laminar flow passes through the specimen, and the knots’ shapes become solid barriers; instead of creating a mesh in the shape of the streamlines, the elementary Cartesian velocity components are recorded. Following that, a solid analysis will be performed, in which solid elements will substitute the fluid components, and the fibre deviation will be calculated using the velocity vector relationships. A fluid-filled upper prismatic pipe surrounds the wood, with knots serving as the fluid’s wall borders.

Furthermore, the effects of the fibre directions and knots on the variation of the MOE were numerically presented aimed for the development of adequate machine strength grading techniques [121]. A three-dimensional finite element model was proposed to study the effects of knots on the strength reduction of Norway spruce timber. It was assumed that the knot cross-section is oval, and its axis could be nonlinear. The orientation in the radial fibre deviation can be represented by the diving angle [122]. The diving angle, also known as an out-of-plane angle, must be addressed when estimating the strength and failure mode of wood specimens; it can approximate the description of fibre orientation out of the plane of the beam face [183]. Moreover, a three-dimensional finite element model was used to predict the mechanical characteristics of Norway Spruce timber boards considering the presence of knots and their corresponding fibre deviations [184].

The knot geometry and associated fibre deviation may also be determined analytically. A mathematical model was suggested to describe the knots and their related fibre deviation boundaries in the three dimensions, assuming that second and third-order polynomial functions may approximate the boundaries. The fibre deviations are determined using the function’s tangent R(*P*, $R_{i0}$); this might be approximated by the distance between the tree’s pith and a particular growth layer near the knot ($R_{i0}$), the distance between the knot’s centre and any examined location (*P*), as well as additional variables relating to the kind of wood member [181].

Knots at the member’s edge are generally the origin of the most substantial stress concentration. The stress concentrations are lowest in knots situated in the middle of the member. The region below the knot is likewise considered to be inefficient and incapable of sustaining stress [185]. A two-dimensional parametric finite element model was used to study the influence of knot size and position on the behaviour of structural wood components. The knots were approximated by openings and cylinders. These models, however, do not include the influence of diving angles [186]. Furthermore, the impact of the knot’s size, location, and dive angle, represented by elliptical oblique and rotating cones, on bending was studied parametrically [187]. The influence of knot size and density on bending strength and flexural stiffness was also investigated [188].

The damage mechanics-based approach assumes that the collapse of timber structures begins with induced ductile micro-defects that develop and spread inside narrow shear bands with highly localized plastic strain. The continuum damage mechanics (CDM) considers orthotropic elastic behaviour, anisotropic plastic damage, isotropic ductile damage, and significant plastic deformation throughout the loading process. Timber elements frequently fail due to transversal tension, and the tensile strength is affected by grain irregularities. The power limiting effects of grain deviations around knots may be described using one-dimensional stress conditions Hankinson’s formula for predicting the longitudinal stresses as a function of the fibre angle in the vicinity of the knot [189]:(22)$$\sigma_{\theta }=\frac{\sigma_{p}\sigma_{q}}{\sigma_{p}\cdot \sin^{2}\theta +\sigma_{q}\cdot \cos^{2}\theta },$$
where $\sigma_{\theta }$ is cross-grain tensile strength at θ, $\sigma_{p}$ is parallel-grain tensile strength, and $\sigma_{q}$ is perpendicular-grain tensile strength. If σ = $\sigma_{o}$, failure occurs.

### 8.2. Knot Represented by Openings

Knots in structural wood are regarded as a significant issue. When detected in the tension zone, they tend to lower the stiffness and ultimate capacity of wood components [190]. Knots were represented as openings in some numerical studies. Ref. [191] studied the local stress concentration numerically due to the presence of knots represented by holes at the tension zones of the beams under bending. They investigated the influence of the size of the hole and its position on the stress distribution in Pinewood which was modelled as a linear elastic orthotropic material. The distribution of normal stresses was identified to be related to the distance from the knot’s centre to the base surface, its diameter, and the amount of loads. As going closer to the neutral axis, the stress ratio between the weakened and non-weakened beams drops. Another research found a decrease in bending stiffness where the knots were also represented by openings, with a 50 mm diameter causing an 83% decrease in bending stiffness [120].

Furthermore, computational and experimental research studies confirmed that openings might replicate the knots. The authors did not observe significant variations in the estimated ultimate forces and strengths (MOR, MOE, maximal force, and deflection) in the timber beams. However, opening would not provide a forewarning of failure. The beams were defected by natural knots of varying location, size, and shape, and then each sample was defected by an opening with a matching fibre pattern. When the load-deflection curves of both approaches were compared, they revealed similar characteristics. The ultimate loads did not differ significantly [192].

A similar study was also carried out. The authors developed an analytical two-dimensional finite element model combined with the Tsai-Wu strength theory for anisotropic material validated with a destructive four-point bending experimental investigation to study the behaviour of recycled timbers containing fastener holes and their effects on the ultimate flexural capacities. They recommended that the distance between the edge of the hole and the extreme compression or tension fibres should not be within 6 mm. The effects of different bolt hole sizes positioned at the midspan of the beam with varying positions relative to the edge of the compression and tension edges on the ultimate flexural capacity of the member were examined. The fastener hole at the beam’s mid-span contributes to the highest section reduction in flexural resistance. A “load-stepping” approach was employed for failure prediction, in which every failed element (reaching 1 as per Tsai-Wu) was removed from the analysis and regarded as a localized failure, resulting in a “net effective section”, which was then utilized in the following analysis step. The load-deflection curves were recorded using the load-step approach until the global failure was reached. For holes that are very close to the edges: 25.4 mm holes with centre 19 mm (6.3 mm far from the edge) and 12.7 mm (touch the edge) far from the tension edge similarly caused the lowest ultimate load capacity. These holes are very close to the tension edge; therefore, a conclusion can be stated that a hole that is very close to the edge has similar effects to that which touches the edge [193].

### 8.3. Knot Represented by a Cone or a Cylinder

On the other hand, knots can be represented by a cone or a cylinder with a diving angle perpendicular to the longitudinal wood fibre. Because the interior geometry of the knot is unknown, assumptions and approximations should be adopted in most situations such that a cone can represent the knot with an inclination concerning the pith of the log. Examples of such approximations are shown by [194]. The ultimate flexural capacity of defected *Pinus sylvestris* L. beams exposed to four-point bending was predicted using finite element modelling assuming different types of knots using an orthotropic elastic-plastic constitutive law for timber in the plane stress state. The rupture region was anticipated based on the position and size of knots as well as grain deviations. Experimentation was also carried out to validate the models [195]. Three types of knots were modelled: a knot represented by an opening, an adherent knot connected to the remainder of the beam, and a partial contact knot with the rest of the beam. The first model predicted the bending failure load for the knots in the tension zone the best. A related work established a three-dimensional analytical material model to predict the behaviour of timber beams while taking into account the impacts of any existing knot in the shape of an elliptical, rotating, or oblique cones [182]. The influence of grain deviation induced by the existence of the knot must be examined since the material characteristics of timber, such as strength and stiffness, are subject to the local fibre orientations. Knots often produce a stress disturbance primarily in their close vicinity, and the associated grain deviations induce a decrease in tensile strength and stress redistribution. Still, their influence on compression strength parallel to the undisturbed grain direction is generally modest.

Table 4 summarizes a few of the investigated studies on the effect of knot size and location on the behaviour of timber members, as well as a quick overview of the simulation concepts of such defects.

## 9. Discussion

This paper comprehensively evaluates experimental, analytical, and numerical efforts to reinforce wood structural components with FRP composites performed by several researchers over the last few decades. Considerable achievements have been made, yet a number of issues remain unresolved.

The most fundamental finding is that synthetic fibres (mainly carbon and glass) constitute a viable and economical reinforcement technique for timber structures, and promising results are obtained with other fibre types as well. The feasibility relies on adequate bond strength requiring proper surface preparations. It is clear that wood material properties significantly influence the behaviour of timber-FRP composite structures and are key factors in understanding its mechanics. Various material models are developed for the stress-strain relationships in timber, including one using different moduli in tension and compression. Moreover, it is found that the pre-stressing reinforcement technology allows the most efficient use of the FRP materials, boosting the load-bearing capacity and providing better design codes.

The authors’ own investigations and the literature review show that the experimentally observed increase of stiffness of reinforced beams under bending often has an inadequate match with theoretical expectations. The authors believe that it is due to the fact that the CFRP materials capacity is not usually fully exploited. This paper presents a one-of-a-kind table based on several papers that compares the increase in stiffness found in various investigations with analytical estimates using the Euler beam model. This table can be used to quickly assess the benefits and draw preliminary conclusions about the predicted outcomes of CFRP materials in the restoration of timber structures. Other tables summarize several of the relevant numerical and experimental results on beam reinforcement, as well as on modelling of knots.

Several topics are still open for future investigations. This study shows the lack of research on defining CFRP-wood interface characteristics, the strengthening of defective timber elements, and the inconsistency of modelling the knots in structural wood. Also, the issue regarding the stiffness mentioned above requires further analysis of the mechanics of timber-FRP composites. A better and detailed understanding of the FRP-wood failure mechanisms, mainly FRP delamination and timber tensile brittle failure, will allow further accurate designs that optimize safety and affordability. Though some of the essential factors influencing the FRP-wood bond were explored, investigations are necessary to have a strong knowledge of the bond’s behaviour. Also, no reports are found in literature on FRP GiR reinforcement potentially introducing some damage to the wood material, therefore, the authors draw the attention to the necessity to research the rod’s effects on wood deterioration. The impacts of knots, including closely-spaced knots, have not been well explored, and there is a need for design codes and criteria for identifying defected wood components. Current approaches for constructing the accurate three-dimensional fibre paradigm of knots and related fibre deviations are limited, and more research is needed.

## Figures and Tables

**Figure 1 polymers-14-02381-f001:**
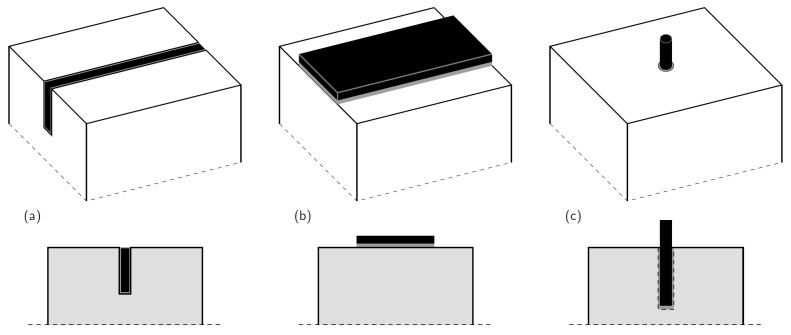
Sketch of strengthening techniques. (**a**) Near-surface mounted (NSM), e.g., at [58], (**b**) Externally bonded reinforcement (EBR), e.g., at [30], (**c**) Glued-in-rods (GiR), e.g., at [59].

**Figure 2 polymers-14-02381-f002:**
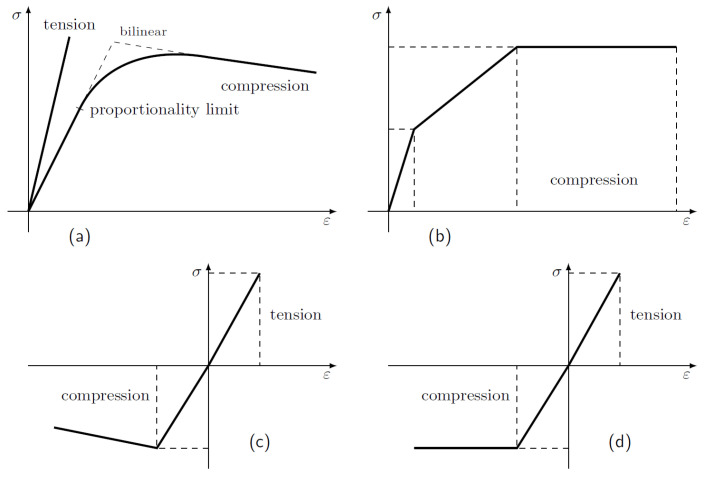
Stress-strain relationship for wood under tension and compression parallel to grain. (**a**) linearity in tension and nonlinearity in compression, e.g., at [46], (**b**) multilinearity in compression, e.g., at [30,37], (**c**) compression parallel to grain as suggested by [104], (**d**) elastic and perfectly plastic in compression, e.g., at [107].

**Figure 3 polymers-14-02381-f003:**
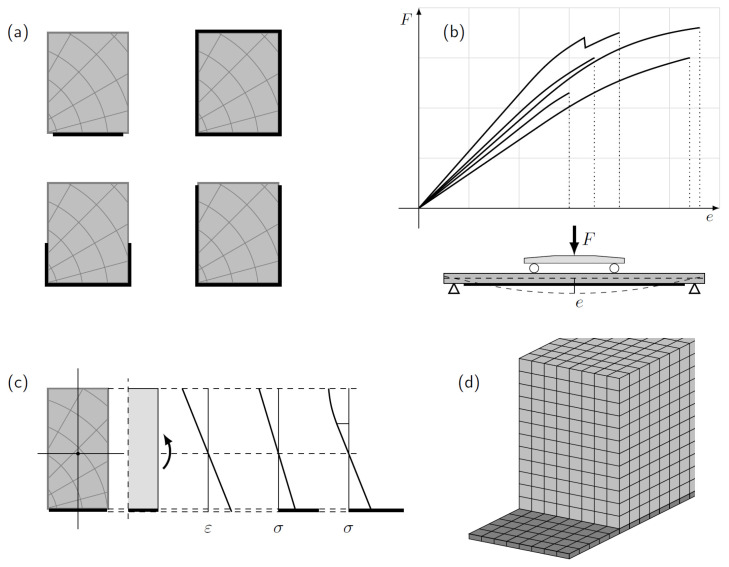
Experimental and numerical analysis of reinforced timber beams. (**a**) Possible cases of externally bonded FRP applied on timber beams, (**b**) common four-point bending test arrangement and typical load-deflection curves in schematic plot, (**c**) uniaxial strain and stress distribution in the classical beam theory considering linearity in tension and linearity or nonlinearity in compression parallel to grain, (**d**) schematic plot of typical 3D finite element model of reinforced beams with block elements.

**Figure 4 polymers-14-02381-f004:**
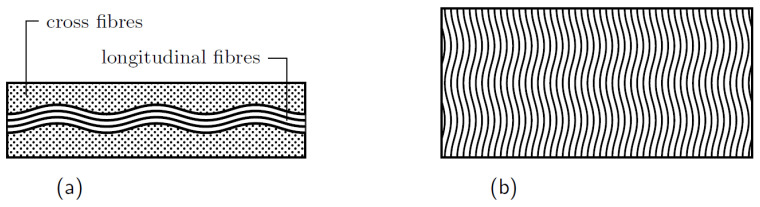
(**a**) Out-of plane fibre waviness in thick composite (section), (**b**) In-plane fibre waviness (top view).

**Figure 5 polymers-14-02381-f005:**
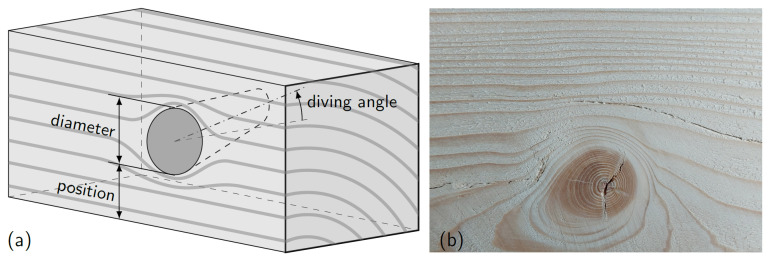
(**a**) Schematic diagram of the vicinity of a knot, (**b**) Test specimen containing a knot after failure; fractures reveal the complex stress distribution around the knot.

**Table 1 polymers-14-02381-t001:** Some of the key experimental, FEM, and theoretical studies. Columns in the table indicate the source, FRP type, composite type, applied prestress, size/amount of FRP, increase of capacity, occurrence of debonding, dominant failure mode, model/analysis, and relevant notes, respectively.

Ref.	FRP Type	Compos.	Pre-Stress	Thickness/Diameter of FRP (mm) or Volume Fraction (%)	Increase of Capacity (%)	Debond.	Dominant Failure Mode (FRP-Wood)	Wood Mechanical Model/Analysis Method	Notes
[37]	GFRP/CFRP	fabrics	no	1% GFRP/0.4% CFRP		no	tensile	brittle elastic in tension, elastic-plastic in compression	surface timber treatment
[130]	CFRP	laminate	no	1.2, 1.4 mm	14–88	yes	shear	elastoplastic; different moduli in tension and compression	beams w/knots
[73]	CFRP	bar	yes	11, 16 mm	93.3–131	no	ductile in compression/flexure–shear	elastoplastic; bilinear in compression	-
[56]	CFRP	laminate	yes/no	1.14 mm	34/22	no	-	linear elastic brittle for tension and ductile for compression	-
[110]	CFRP	laminate	no	1.2 mm	28	no	crack initiation at the reinforcement boundaries	linear elastic brittle for tension and ductile for compression	interfacial shear and peeling stresses are essential for failure analysis
[60]	GFRP	laminate	yes	3.3 mm	95	no	tensile	elastic-plastic in compression/linear elastic in tension-moment-curvature analyses	During the wet bond line, the approach was to release the pre-tensioning force
[100]	CFRP	rod/laminate	no	-	49–63	no	ductile compression	-	anchoring length provided to avoid delamination
[107]	CFRP	laminate	no	2.8 mm	28.6–38	no	tensile	-	a nonlinear numerical model was developed to accurately replicate wood behaviour
[66]	CFRP	laminate	no	1.2 mm	21–79	no	tensile (mainly in the defected timber zones)	elastic-ideally plastic material model	inserting the CFRP inside the cross-section reduces the possibility of delamination failure

**Table 2 polymers-14-02381-t002:** CFRP versus GFRP.

In Terms of	CFRP	GFRP
definition	Carbon fibre Reinforced polymers	Glass fibre Reinforced Polymers
properties	lightweight (1.7 g/mm${}^{3}$)	medium weight (2.5 g/mm${}^{3}$)
conductivity	conductive	insulative
cost	expensive	acceptable price
fibre diameter	fine	thicker
properties	orthotropic (simplified to quasi-isotropic)	can be simplified to isotropic

**Table 3 polymers-14-02381-t003:** The increase in stiffness in some studies versus the theoretical increase based on Euler-beam model. $E_{w}$ modulus of elasticity for wood (GPa); $E_{f}$ modulus of elasticity for CFRP (GPa); wood dimensions as height × width × length (mm); CFRP dimensions as width × thickness (mm); C and T stand for compression and tension, respectively.

References	CFRP Type	$E_{w}$	$E_{f}$	FRP Dimensions	Wood Dimensions	Stiffness Increase (%)	Theoretical Stiffness Increase (%)
[37]	fabrics	C:10.5/T:8	180	Full width of the beam.	60 × 120 × 3000	15–30	25.228
[110]	strips	15.3	161	50 × 1.2	50 × 50	23–28	63.427
[100]	strips	10.4	155	10 × 10	40 × 40 × 40	10	68.548
[155]	bidirectional fabrics	13.238	137.895	-	483 × 203 × 7620	17–27	18.457
[109]	sheets	11	230	100 × 0.165	145 × 115	15–60	6.095
[161]	sheets and plates	13	165	25 × 1.5	50 × 25 × 500	20.2–29.6	87.768
[38]	sheets	C: 8.17/T:8.68	100.19	40 × 1	60 × 40 × 900	36.19–64.12	49.793
[108]	sheets	-	417.625	100 × 0.165	200 × 200 × 4000	22.5–30.3	-
[39]	sheets and strips	9.9	227	35 × 0.165	138 × 38 × 2440	25–50	7.406
[46]	strips	9	165	43 × 1.2	160 × 43 × 1650	40	36.810
[45]	plates	8	45	96 × 2.8	96 × 44 × 4200	13.15	83.760
[107]	plates	8	38.44	96 × 3.3	190 × 96 × 1710	8.4	23.918
[157]	lamella	13	164	1.4 × 1	15.41 × 15.41 × 3150	6	23.551
[36]	fabrics	Oak:9/Fir:10	417.6	20 × 0.165	20 × 20 × 380	4.7–15.1	84.425
				67 × 0.165	67 × 67 × 1320		30.917
				100 × 0.165	100 × 100 × 1950		21.405
				100 × 0.165	200 × 200 × 4000		5.643

**Table 4 polymers-14-02381-t004:** Experimental and numerical studies on the effects of knots in structural timber.

Ref.	Knot Approximated by	Knot Location	Knot Diameter (mm)	Loss in Capacity (%)	Notes
[186]	opening/solid	variable throughout the beam height	variable	1–76	Opening model was a good assumption for the knots located in the tension zone of a beam under bending.
[192]	opening/solid	-	-	-	Knots were found to be similar to artificial openings in their effects on the MOR.
[177]	solid	variable	variable	-	Two closely spaced knots can diminish tensile strength by 60% when compared to a single critical knot.
[120]	opening	mid-span in the tension zone	variable	-	A 50 mm diameter opening may reduce the bending stiffness by 83%.
[195]	opening/solid	tension zone	28.5–43.7	-	The concept employs knots as an opening for those in the tension zone is practical.
[182]	ellipse, rotated and oblique cone	tension zone	variable	-	The 3D flow-grain analogy was applied to determine the grain deviation around the knot.
[187]	cylindrical, truncated conical, shallow conical face knots and conical edge knots	variable	variable	72, due to size: 55—due to position: 70, due to size: 43—due to position: 68, due to size: 52—due to position: 24	The knot inclination can sometimes increase the flexural capacity of the timber element.
[184]	cone	variable	variable	-	The precision of knot modelling was demonstrated to be dependent on accurate pith modelling.
[190]	opening/solid	variable	30-40	up to 62	The findings highlight the need of correct modelling for fibre deviations rather than the knot itself.

## Data Availability

Not applicable.

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
