# Peer review of "Strengthening Timber Structural Members with CFRP and GFRP: A State-of-the-Art Review"

_polymers, 2022, doi:10.3390/polym14122381_

Round 1

Reviewer 1 Report

This research paper is about the CFRP and GFRP studies reviewing the strengthening timber beams.

  1. This review is based on many previous classical reviews and the summary of relevant reviews is important to be included to reflect the technical merits.
  2. section 4 strengthening mechanism is interesting but a theoretical summary considering some relevant simulations will be nice, including finite element studies devoted a lot to the composite mechanics understanding. 
  3. would the interfacial bonding not limited to the waviness but also the fiber surface treatment help the strengthening mechanism? especially considering the large body of literature in that area.

Reviewer 2 Report

The current paper provides a comprehensive and systematic review on FRP strengthened wood structures. Overall, the paper is significant for the readers, which provides a good research direction. Some further improvements still need to be made by referring to the specific comments below.

*Abstract,

As a review paper, the abstract should include a systematic summary of the latest research achievement and some key conclusions should be further obtained about FRP strengthened wood structures, such as loading bearing capacity, stress distribution, failure mode and mechanism and so on. In addition, some suggestions for future research work for the unsolved problems should be emphasized according to the research status.

1. Aim and scope of pursued study

1) In this part, when summarizing the research status, relevant literature sources should be added. For fiber reinforced polymer composites (FRP), the reviewer suggests the authors providing the composition (fiber, polymer and interface) and type, such as natural fiber reinforced polymer (NFRP) composites and synthetic fiber reinforced polymer (CFRP, GFRP and BFRP) composites. By comparing the above two kinds of FRP, the advantages of synthetic fiber composites compared with natural fiber composites and the applicability of engineering strengthening should be put forward. In addition, the successful application of CFRP and GFRP used in the engineering strengthening is attributed to their lightweight, high strength, excellent corrosion/creep and fatigue resistances. Please see the latest research on the performance and advantages of CFRP and GFRP composites compared to NFRP.

Superior mechanical properties and creep resistance: Composite Structures, 2022. 281: 115060.  NFRP: Materials Today: Proceedings, 2020, 28: 1616-1621.  Durability resistance: Construction and Building Materials, 2022, 315: 125710.

2) Line 39-41, “The high strain capacity of CFRP reinforcement allows wood compression fibres to yield and better exploit flexural capacity.”, the above sentence is easy to cause ambiguity. This is because CFRP itself is a brittle material, it has low ultimate elongation at break (<2%).

3) For the seven topics under discussion, there are some content repetitions and unclear logical logicality. For example, parts 4 and 5 should be included in part 3. Part 7 does not specify CFRP or GFRP, which can not be related with other contents. It is suggested that the authors should make further adjustments so as to avoid duplication of content and increase the logic of discussion.

  1. Quick historical overview

For the historical overview on the strengthening of wood components, it is suggested to conduct further analysis and summary according to the years of time. Therefore, the content of paragraph one is suggested to be placed in the last paragraph. In addition, after 1992, are there any other new composite used to strengthen the wood structure?

  1. General introduction

In this part, it is suggested that the authors adjust the current writing to increase the clear logic based on material (timber, CFRP and GFRP), reinforcement technology/method, and the performance evaluation and investigation (experiment and finite element simulation) after the strengthening.

  1. CFRP strengthening techniques

1) Figure 1 shows the schematic diagram of three main strengthening techniques. As known, compared with the externally bonded reinforcement, the prestressed reinforcement technology is a very effective method through the anchoring effect at both ends of FRP. It can greatly improve the bearing capacity of strengthened wood structures, delay the cracking, and so on. Some research status of the above new reinforcement technology should be also analyzed.

2) In the part of 4.3 (Glued-in rods (GiR) strengthening technique, it seems that the technique has destroyed the wood structure. Will this cause some damage to the raw materials themselves?

  1. CFRP-wood bond characteristics

1) The premature failure due to debonding can be well avoided by the prestressed reinforcement technology.

2) In order to improve the interface bonding performance between FRP and wood structure, the surface treatment method may be very important. For example, the polishing of FRP and wood surface, the high-performance modification treatment of adhesives etc.

  1. Numerical and experimental studies on CFRP-wood strengthening

1) This part is basically summarized in the form of words. It is suggested that the authors present the results of some key experiments and simulations in the form of graphs and tables. At the same time, some key parameters affecting the reinforcement performance should be analyzed emphatically.

  1. Modelling of knots in structural timber

1) The simulation results of structural timber joints should be expressed quantitatively through the figures and tables.

  1. Discussion

1) The authors should add the conclusion and prospect analysis. For the conclusion, it should present some important research results on FRP reinforced wood structures. The prospect analysis should further put forward the problems to be solved in the future research.

Reviewer 3 Report

Abstract

-Strengthening timber beams or Strengthening wood ?

-Please revise “The topic got so wide and diverse, and many contradicting results emerged that
a thorough review has become necessary”

-Please remove “Seven chapters covering the main topics in the field of
FRP-wood strengthening are presented”

-Please revise “The authors draw attention to several challenges (e.g. moderate stiffness enhancement). It can be a starting point for future research and engineering projects.”

-Please highlight the significance of CFRP and GFRP in strengthening timber beams in the conclusions for what applications etc. ?

-Authors may conclude words from line 8 to 14 by 2 to 3 sentence.

7. Effect of the CFRP on flexural stiffness enhancement of timber

-Please add another column in Table 1 to specify what kind of fibre they are used.

General comments

-I suggest authors to read and restructure the positions of each section, its confusing.

-Also, please re-structure the lengthiness for 1 paragraph...its too long for a certain paragraph to read and to stop.

Round 2

Reviewer 2 Report

The paper can be accepted in present form.